# Teleport Graph Convolutional Networks

## Abstract

We consider the limitations in message-passing graph neural networks. In message-passing operations, each node aggregates information from its neighboring nodes. To enlarge the receptive field, graph neural networks need to stack multiple message-passing graph convolution layers, which leads to the over-fitting issue and over-smoothing issue. To address these limitations, we propose a teleport graph convolution layer (TeleGCL) that uses teleport functions to enable each node to aggregate information from a much larger neighborhood. For each node, teleport functions select relevant nodes beyond the local neighborhood, thereby resulting in a larger receptive field. To apply our structure-aware teleport function, we propose a novel method to construct structural features for nodes in the graph. Based on our TeleGCL, we build a family of teleport graph convolutional networks. The empirical results on graph and node classification tasks demonstrate the effectiveness of our proposed methods.

## 1 Introduction

Graph neural networks (GNNs) have shown great capability in solving challenging tasks on graph data such as node classification (Grover & Leskovec, 2016; Kipf & Welling, 2017; Veličković et al., 2017; Gao et al., 2018), graph classification (Xu et al., 2018; Gao & Ji, 2019; You et al., 2019), and link prediction (Zhang & Chen, 2018; Chen et al., 2019; Zhou et al., 2019). Most graph convolutional networks are based on message-passing operations, in which each node aggregates information from its neighboring nodes. To enable a larger receptive field (Chen et al., 2016), GNNs need to stack multiple layers, which is straightforward but can result in several issues. Firstly, stacking multiple layers involves massive trainable parameters, which consequently increases the risk of over-fitting. Secondly, message-passing operations mostly use averaging to combine the aggregated features, which significantly reduces the distinguishability of network embeddings. From this point, GNNs that are based on message-passing operations can not use deep network architecture due to these limitations. Some works such as Geom-GCN (Pei et al., 2020) try to solve these issues by involving more nodes in the feature aggregation process. However, Geom-GCN doesn't consider the original graph topology information when generating the additional set of nodes for aggregation, which can neglect some relevant nodes from a structural perspective.

To address the above limitations and increase the receptive field effectively, we propose a teleport graph convolution layer (TeleGCL) that uses teleport functions to select highly-relevant nodes at the global scope. A teleport function computes relevances between the center node and other nodes beyond the local neighborhood. The nodes with particular relevances are teleported for the center node. Here, the selection of teleported nodes is not restricted by the graph topology. This enables the center node to gather information from a larger neighborhood without going deep, which helps to avoid over-fitting and over-smoothing issues. In particular, we propose two teleport functions; those are structure-aware and feature-aware teleport functions. They compute the nodes' relevances from graph structural perspective and node features perspective, respectively. Based on our TeleGCL, we build a family of teleport graph convolutional networks. The empirical results on graph and node classification tasks demonstrate the effectiveness of our proposed methods.

## 2 Background and Related Work

In this section, we describe message-passing operations on graph data and geometric graph convolutional networks. Graph neural networks (Fan et al., 2019; Wu et al., 2019; Morris et al., 2019; Wu

et al., 2020) have achieved state-of-the-art performances on various challenging tasks in the field of network embedding. The mainstream of graph deep learning operations follows a message-passing schema. In a message-passing operation, each node sends its features, known as message, to its neighboring nodes in the graph. Then each node aggregates messages from its neighborhood and uses them to update its features. When combing the aggregated features, different strategies can be applied. In the graph convolution layer (GCN) (Kipf & Welling, 2017), features from neighboring nodes are given equal weights in the aggregation process. To assign different weights to different neighboring nodes, the graph attention network (Veličković et al., 2017) employs an attention mechanism to compute aggregation weights. Based on these message-passing operations, graph neural networks stack multiple layers, which enables a larger receptive field. Recently, some research works try to perform message passing beyond the local neighborhood. Pei et al. (2020) proposed to construct a continuous latent space that enables graph neural networks to perform feature learning in the latent space. To be specific, it first projects nodes' features to a 2-dimensional latent and continuous space. Based on the latent space, a structural neighborhood is constructed based on the Euclidean distance of each pair of nodes in the 2-dimensional space. In this process, the construction of structural features does not consider the graph connectivity information in the graph. Thus, the structural neighborhood in (Pei et al., 2020) is still built on node features without considering the graph topology. In this work, we propose a method to generate structure-aware features for each node. In particular, we use the graph connectivity and similarity information with the neighboring nodes and construct a feature vector for each node. By considering graph connectivity, our constructed structural features can reflect graph topology information.

# 3 TELEPORT GRAPH CONVOLUTIONAL NETWORKS

In this work, we propose the teleport graph convolution layer (TeleGCL) that enables a center node to aggregate information beyond regular neighborhood structure by using some teleport functions. To enable effective node teleportation, we propose two teleport functions from structure-aware and feature-aware perspectives. Specifically, we propose a novel method to construct structural features for nodes, which can be used by structure-aware functions to select relevant nodes. Based on our TeleGCL, we propose the teleport graph convolutional networks for network embedding learning.

## 3.1 LIMITATIONS OF MESSAGE-PASSING OPERATIONS

Currently, most graph convolution networks are based on message-passing operations. In a message-passing operation, each node aggregates information from its neighboring nodes that usually are the one-hop neighborhood. Intuitively, it is beneficial to use information from a large neighborhood for network embedding learning. To enlarge the receptive field, a straight way is to stack multiple message-passing layers. A graph convolutional network with $k$ layers enables nodes to receive information from a $k$-hop neighborhood. However, this method results in two issues. Firstly, it increases the risk of over-fitting by involving much more trainable parameters. The number of trainable parameters in the network increases when stacking multiple layers. Unlike regular convolutional neural networks, there is no effective graph pooling layer that can enlarge the receptive field without involving trainable parameters. Stacking many graph convolution layers will inevitably increase the risk of over-fitting. Secondly, stacking multiple layers will reduce the distinguishability of network embeddings, which is often referred to as the over-smoothing issue (Pei et al., 2020). Due to the invariant property in graph structures, message-passing operations cannot learn trainable weights in the aggregation process (Kipf & Welling, 2017; Gao et al., 2018). Averaging operation is usually used for information aggregation from the neighborhood. Consequently, information from relevant distant nodes will be diluted and each node carries similar information. In this work, we propose a teleport graph convolution layer to address this issue. This layer enables each node to aggregate information from a set of relevant nodes that are not directly connected to the center node in the original graph structure. Teleport functions are used to determine the relevant nodes from different perspectives.

## 3.2 TELEPORT GRAPH CONVOLUTION LAYER

To address the limitations in message-passing operations, we propose the teleport graph convolution layer (TeleGCL), which enables nodes to aggregate information beyond their local neighborhoods.

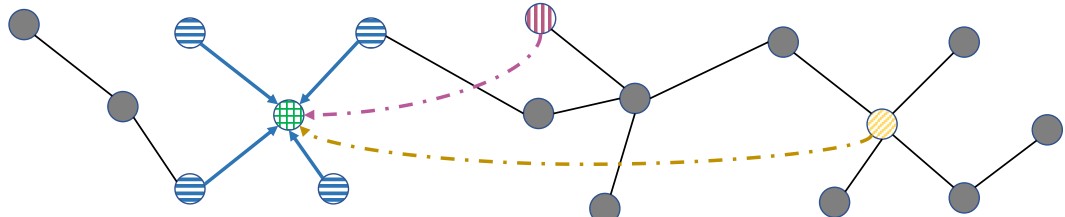

Figure 1: Illustration of our proposed teleport graph convolution layer. In this layer, the center node (green node) aggregates information from its local neighborhood (blue nodes), a feature-aware neighborhood (purple nodes), and a structure-aware neighborhood (yellow nodes).

In this layer, we employ multiple teleport functions to generate neighborhoods for each node. The teleport functions select some nodes that are relevant but not directly connected. Since these nodes are teleported from a global context, the receptive field of each node can be effectively enlarged. We require these functions to be permutation invariant such that the property retains in this layer.

Given a graph $\mathbb{G} = (V, E)$, where $n = |V|$, each node $v \in V$ is associated with a feature vector $\boldsymbol{x}_v \in \mathbb{R}^d$, and each edge $(u, v) \in E$ connects node $u$ and node $v$ in the graph. $\boldsymbol{X} = [\boldsymbol{x}_1, \boldsymbol{x}_2, \ldots, \boldsymbol{x}_n] \in \mathbb{R}^{d \times n}$ and $\boldsymbol{A} \in \mathbb{R}^{n \times n}$ are the feature matrix and adjacency matrix, respectively. A teleport function $g(v, \mathbb{G}) \to \mathbb{N}$ takes as input a node $v$ and outputs a neighborhood $\mathbb{N}$ that includes a set of relevant nodes for node $v$'s feature aggregation. In Section 3.3 and Section 3.4, we propose two teleport functions that can construct structure-aware and feature-aware neighborhoods. Suppose we have $m$ teleport functions, node $v$ aggregates information from a neighborhood $\mathbb{N}(v) = \{\mathbb{N}_l(v), \mathbb{N}_1(v), \mathbb{N}_2(v), \ldots, \mathbb{N}_m(v)\}$, where $\mathbb{N}_l(v) = \{u | u \in V, (v, u) \in E\}$ is the local neighborhood and $\mathbb{N}_i(v) = g_i(v, \mathbb{G})$ is a neighborhood created by the $i^{th}$ teleport function. Based on this neighborhood, the layer-wise propagation of TeleGCL $\ell$ for node $v$ is formulated as

$$\boldsymbol{x}_v^{(\ell)} = \sigma \left( \frac{1}{|\mathbb{N}(v)|} \sum_{u \in \mathbb{N}(v)} \boldsymbol{x}_u^{(\ell-1)} \right), \tag{1}$$

where $\sigma$ denotes an activation function such as ReLU (Nair & Hinton, 2010). By using teleport functions, a node can aggregate features beyond the local neighborhood, thereby leading to a larger receptive field.

In Section 3.2, we propose the teleport graph convolution layer that enables feature aggregation regardless of the original graph structure. A TeleGCL highly depends on teleport functions which select distant nodes in the feature aggregation process. In previous works (Pei et al., 2020), teleport functions are mainly based on node features while neglecting the graph structural information. The graph topology information should be considered to include nodes that share the same structural patterns such as graph motifs (Ren & Jin, 2020). In this section, we propose two teleport functions; those are structure-aware teleport function and feature-aware teleport function. They select teleport nodes based on graph topology and node features, respectively.

### 3.3 STRUCTURE-AWARE TELEPORT FUNCTION

Structure-aware teleport function focuses on selecting nodes based on the graph topology. In a graph, the nodes that share the same structural pattern contain related features. It is desirable for a node to aggregate information from these relevant nodes. A structure-aware function can be used to capture relevant nodes from a graph structural perspective. With a structure-aware function, the teleported nodes for node $v$ are selected as:

$$\mathbb{N}_s(v) = \{u | u \in V, (v, u) \notin E, t_s(v, u) > \theta_1\}, \tag{2}$$

where $t_s(v, u)$ is a structure-aware function that computes the relevance of two nodes from a structural perspective. Here, $\theta_1$ is used to determine if node $u$ is relevant.

In this work, we propose a novel structure-aware teleport function, which computes the relevance of two nodes by checking if they share the same structural pattern. Our proposed method is based

Figure 2: Illustration of our proposed method to construct structural features for a node. Given a center node (green node), we first compute similarity scores between it and each of its neighboring nodes. Then, we rank these scores and select the $k$-largest values to form its structural feature vector.

on an intuition that the nodes with the same structural pattern have similar connections with their neighboring nodes. From this point, we create a structure feature vector for each node, which can reflect its structural pattern such as graph motifs. For each node, we first compute its similarity scores with its neighboring nodes. Then we rank these similarity scores and use them as the structural feature of this node. To be specific, the structural feature vector $\boldsymbol{y}_v$ for node $v$ is constructed as

$$\boldsymbol{w}_v = \boldsymbol{X}^T \boldsymbol{x}_v, \qquad\qquad \in \mathbb{R}^n \qquad (3)$$

$$\boldsymbol{idx}_v = \mathrm{rank}_k \left( \boldsymbol{w}_v \circ \boldsymbol{A}_{:,v} \right), \qquad\qquad \in \mathbb{R}^k \qquad (4)$$

$$\boldsymbol{y}_v = \boldsymbol{w}_v \left( \boldsymbol{idx}_v \right), \qquad\qquad \in \mathbb{R}^k \qquad (5)$$

where $\circ$ denotes an element-wise vector multiplication, and $\boldsymbol{A}_{:,v}$ is the $v^{th}$ column of the adjacency matrix. $\mathrm{rank}_k$ operator ranks the similarity scores and outputs the indices of the top-$k$ values in $\boldsymbol{w}_v$. $\boldsymbol{w}_v(\boldsymbol{idx}_v)$ returns a subset of rows in $\boldsymbol{w}_v$ indexed by $\boldsymbol{idx}_v$.

We first compute the similarity scores between node $v$ and its neighboring nodes in Eq. (3). Each element $w_{u,v}$ in $\boldsymbol{w}_v$ measures the similarity between node $u$ and node $v$. In Eq. (4), we rank these similarity scores and select the $k$-largest values in $\boldsymbol{w}_v$. The indices of the selected values are stored in $\boldsymbol{idx_v}$. Using indices $\boldsymbol{idx_v}$, we extract a structural feature vector $\boldsymbol{y}_v$ from $\boldsymbol{w}_v$. By repeating these operations on each node, we can obtain a structural feature matrix $\boldsymbol{Y} = [\boldsymbol{y}_1, \boldsymbol{y}_2, \ldots, \boldsymbol{y}_n] \in \mathbb{R}^{k \times n}$ for all nodes in the graph. In this way, the structural feature vector is constructed from similarity scores between the center node and its neighboring nodes. These similarity scores encode its connectivity pattern with surrounding nodes, thereby reflecting the structural information in the graph.

Based on structural features, we use dot product to compute the relevance of node $u$ and node $v$: $t_s(u, v) = \mathrm{softmax}(\boldsymbol{y}_u^T \boldsymbol{y}_v)$, which can measure relevance from the perspectives of both angle and magnitude in an efficient way. As illustrated in Eq. (2), the teleport nodes can be selected based on our constructed structural features.

### 3.4 FEATURE-AWARE TELEPORT FUNCTION

In a feature-aware teleport function, the teleported nodes are selected based on node features. A feature-aware teleport function can select highly relevant nodes based on their features. By using this function, the teleported nodes for node $v$ are selected as:

$$\mathcal{N}_f(v) = \{u | u \in V, (v, u) \notin E, t_f(v, u) > \theta_2\}, \qquad (6)$$

where $t_f(v, u)$ is a teleport function. $\theta_2$ is a threshold to determine if a node is teleported.

Notably, Geom-GCN (Pei et al., 2020) uses a special case of this feature-aware teleport function. In Geom-GCN, node features are projected into a 2-dimensional space then the Euclidean distance is computed and used. The structural features in Geom-GCN are based on the latent space without considering graph topology information. The time complexity of this function is $O(2d)$.

In our feature-aware teleport function, we use dot product to compute the relevance, which is effective and can slightly reduce the computational cost. To be specific, the feature-based relevance between node $u$ and node $v$ is computed as $t_f(v, u) = \mathrm{softmax}(\boldsymbol{x}_u^T \boldsymbol{x}_v)$. By combining structure-aware and feature-aware teleport functions, the neighborhood for node $v$ is defined as $\mathcal{N}(v) = \{\mathcal{N}_l(v), \mathcal{N}_f(v), \mathcal{N}_s(v)\}$. In our proposed TeleGCL, each node aggregates information from nodes in neighborhood $\mathcal{N}(v)$.

Figure 3: Illustration of our proposed teleport graph convolutional networks. In this example, the input graph contains five nodes, each of which has two features. We first use a GCN layer to learn a new feature embedding for each node. Each of the following two blocks contains a TeleGCL and a pooling layer to reduce graph size. The outputs of all convolution layers are globally reduced and fed into the final multi-layer perceptron for prediction.

## 3.5 TELEPORT GRAPH CONVOLUTIONAL NETWORKS

Based on our TeleGCL, we build a family of teleport graph convolutional networks (TeleGCNs). Given an input graph, we first use an embedding layer to learn low-dimensional continuous feature embeddings for nodes in the graph. Possible choices for this embedding layer include fully-connected layer and GCN layer. Here, we use a GCN layer to learn feature embeddings. Then several convolutional blocks are stacked to gradually learn network embeddings. In each convolutional block, we use our TeleGCL to learn high-level feature embedding, and a pooling layer to reduce the graph size and involve more non-linearity. Here, we use a sampling-based pooling method to retrain original graph structures and reduce the risk of over-fitting. Specifically, we use top-$k$ pooling (Gao & Ji, 2019) layers in our model. Finally, we stack the outputs of all TeleGCLs and the output of the first GCN layer. To deal with variable graph sizes in terms of the number of nodes in a graph, we employ several global pooling layers such as averaging, maximization, and summation to reduce these outputs into vectors. These feature vectors are concatenated and fed into a multi-layer perceptron (MLP) for prediction.

## 4 EXPERIMENTS

In this section, we conduct experiments on graph classification tasks to evaluate our proposed methods. We compare our teleport graph convolutional networks (TeleGCNs) with previous state-of-the-art models. We conduct ablation studies to investigate the contributions of our proposed teleport functions. Some experiments are performed to study the impact of thresholds in teleport functions. Our code and detailed experimental setups are available in the supplementary material.

## 4.1 RESULTS ON GRAPH CLASSIFICATION TASKS

We evaluate our proposed TeleGCL and TeleGCNs on graph classification tasks. We compare our TeleGCNs with the previous model on seven datasets including PROTEINS (Borgwardt et al., 2005), COLLAB, D&D (Dobson & Doig, 2003), IMDB-MULTI (Yanardag & Vishwanathan, 2015a), REDDIT-BINARY, REDDIT-MULTI-5K, and REDDIT-MULTI-12K (Yanardag & Vishwanathan, 2015b). These datasets are benchmarking graph datasets and are widely used for evaluation in this community. Notably, these datasets have no test dataset. The common practice (Xu et al., 2018; Ying et al., 2018; Gao & Ji, 2019; Lee et al., 2019) is to run 10-fold cross-validation on the training dataset and report the average accuracy (%) with standard deviation. We choose six previous state-of-the-art models as baseline models (Shervashidze et al., 2011; Niepert et al., 2016). We strictly follow the same practices as previous works. In bioinformatics datasets such as PROTEINS and D&D, we use original node features in the datasets. In social network datasets like REDDIT-BINARY and IMDB-MULTI, we use node degrees as their initial features. The hyper-parameters in TeleGCNs are slightly tuned on D&D dataset and are migrated to other datasets with slightly different selections.

The experimental results on graph classification tasks are summarized in Table 1. Here, we report the graph classification accuracies with standard deviations. It can be seen from the results that our proposed TeleGCNs achieve significantly better performances than previous state-of-the-art models on six out of seven datasets. To be specific, our TeleGCNs outperform previous models by margins of 1.4%, 2.4%, 21.7%, 4.6%, 1.9%, and 5.6% on PROTEINS, COLLAB, D&D, IMDB-MULTI,

Table 1: Results on graph classification tasks using PROTEINS, COLLAB, D&D, IMDB-MULTI, REDDIT-BINARY, REDDIT-MULTI-5K, and REDDIT-MULTI-12K datasets. We compare our TeleGCNs with previous state-of-the-art models. We report the graph classification accuracies (%) with standard deviations on these datasets.

| | PROTEINS | COLLAB | D&D | IMDBM | REDDITB | REDDIT5 | REDDIT12 |
|---|---|---|---|---|---|---|---|
| *#classes* | 2 | 3 | 2 | 3 | 2 | 5 | 11 |
| *#graphs* | 1113 | 5000 | 1178 | 1500 | 2000 | 4999 | 11929 |
| *#nodes* | 39.1 | 74.5 | 284.3 | 13 | 429.6 | 508.5 | 391.4 |
| WL | $75.0 \pm 3.1$ | $78.9 \pm 1.9$ | $78.3 \pm 0.6$ | $50.9 \pm 3.8$ | $81.0 \pm 3.1$ | $52.5 \pm 2.1$ | $44.4 \pm 2.1$ |
| PSCN | $75.9 \pm 2.8$ | $72.6 \pm 2.2$ | $76.3 \pm 2.6$ | $45.2 \pm 2.8$ | $86.3 \pm 1.6$ | $49.1 \pm 0.7$ | $41.3 \pm 0.8$ |
| DIFFPOOL | 76.3 | 75.5 | 80.6 | - | - | - | 47.1 |
| DGCNN | $75.5 \pm 0.9$ | $73.8 \pm 0.5$ | $79.4 \pm 0.9$ | $47.8 \pm 0.9$ | - | - | $41.8 \pm 0.6$ |
| SAGPool | 71.9 | - | 76.5 | - | - | - | - |
| Top-$k$ Pool | $77.6 \pm 2.6$ | $77.5 \pm 2.1$ | $82.4 \pm 2.9$ | $51.8 \pm 3.7$ | $85.5 \pm 1.3$ | $48.2 \pm 0.8$ | $44.5 \pm 0.6$ |
| GIN | $76.2 \pm 2.8$ | $80.6 \pm 1.9$ | $82.0 \pm 2.7$ | $52.3 \pm 2.8$ | $92.4 \pm 2.5$ | $\mathbf{57.5 \pm 1.5}$ | - |
| **TeleGCN** | $\mathbf{79.0 \pm 4.3}$ | $\mathbf{83.0 \pm 1.3}$ | $\mathbf{84.1 \pm 2.9}$ | $\mathbf{56.9 \pm 3.2}$ | $\mathbf{94.3 \pm 0.7}$ | $56.6 \pm 2.1$ | $\mathbf{50.1 \pm 1.3}$ |

REDDIT-BINARY, and REDDIT-MULTI-12K datasets. The results above show that our TeleGCNs consistently yield state-of-the-art performances on graph classification tasks, which demonstrate the effectiveness of our methods. By using teleport functions, TeleGCL can rapidly and effectively increase the receptive fields without involving massive trainable parameters.

## 4.2 PERFORMANCE STUDY ON SMALL DATASETS

The experimental studies in Section 4.1 on relatively large datasets in terms of the number of graphs demonstrate the effectiveness of our proposed methods. In this section, we conduct experiments to study the performances of our TeleGCNs on three relatively small datasets; those are MUTAG (Wale et al., 2008), PTC (Toivonen et al., 2003), and IMDB-BINARY (Yanardag & Vishwanathan, 2015a). MUTAG and PTC are bioinformatics datasets, while IMDB-BINARY is a popular social network dataset. We follow the same experimental setups as previous works (Xu et al., 2018; Ying et al., 2018; Gao & Ji, 2019). The experimental setups for experiments are provided in the supplementary material. The experimental results

Table 2: Results on graph classification tasks using MUTAG, PTC, and IMDB-BINARY datasets. We compare our TeleGCNs with previous state-of-the-art models. We report the graph classification accuracies with standard deviations.

| | MUTAG | PTC | IMDBB |
|---|---|---|---|
| *#classes* | 2 | 2 | 2 |
| *#graphs* | 188 | 344 | 1000 |
| *#nodes* | 17.9 | 25.5 | 19.8 |
| WL | $90.4 \pm 5.7$ | $59.9 \pm 4.3$ | $73.8 \pm 3.9$ |
| PSCN | $92.6 \pm 4.2$ | $60.0 \pm 4.8$ | $71.0 \pm 2.2$ |
| DGCNN | $85.8 \pm 1.7$ | $58.6 \pm 2.4$ | $70.0 \pm 0.9$ |
| Top-$k$ Pool | $87.2 \pm 7.8$ | $64.7 \pm 6.8$ | $75.4 \pm 3.0$ |
| GIN | $90.0 \pm 8.8$ | $64.6 \pm 7.0$ | $75.1 \pm 5.1$ |
| **TeleGCN** | $\mathbf{93.5 \pm 6.8}$ | $\mathbf{73.8 \pm 3.7}$ | $\mathbf{79.8 \pm 2.5}$ |

are summarized in Table 2. Due to the lack of testing datasets, we follow the common practices in previous works (Xu et al., 2018; Ying et al., 2018) and report the average accuracies by running 10-fold cross-validation on the training dataset. It can be seen from the results that our TeleGCNs achieve promising results on these relatively small datasets. Our TeleGCNs outperform previous models by margins of 0.9%, 9.1%, and 4.4% on MUTAG, PTC, and IMDB-BINARY datasets. The good performances on small datasets demonstrate that our proposed TeleGCL can effectively increase the receptive field without increasing the risk of over-fitting. To be specific, TeleGCL achieves a larger receptive field by using teleport functions, which teleport relevant nodes without using extra trainable parameters.

## 4.3 ABLATION STUDY OF TELEPORT FUNCTIONS

In this section, we conduct experiments to study the contributions of our proposed teleport functions to the overall performances. In our TeleGCL, we use both structure-aware and feature-aware teleport functions to enlarge the receptive fields without stacking many graph convolution layers. The promising results in previous sections have demonstrated the effectiveness of our methods. To investigate the individual contributions of each teleport function, we build multiple networks with

the same network architecture as TeleGCN. To be specific, we build two networks that only use the feature-aware and structure-aware teleport functions. We denote these two networks as TeleGCN-$t_s$ and TeleGCN-$t_f$, respectively. Also, we replace our TeleGCLs with GCNs in the network, which results in GCNet. We evaluate these networks on IMDB-MULTI, REDDIT-BINARY, and REDDIT-MULTI-12K datasets. To ensure fair comparisons, we use the same experimental setups for these networks. The experimental results are summarized in Table 3. It can be seen from the results that the best performances are achieved when two teleport functions are used. The networks

Table 3: Comparison of our TeleGCNs with the networks using the same network architecture with TeleGCL replaced by GCN (denoted as GCNet) and the networks only with structure-aware teleport function (denoted as TeleGCN-$t_s$) and that with feature-aware teleport function (denoted as TeleGCN-$t_f$), respectively. We report the graph classification accuracies with standard deviations on three datasets.

|  | IMDBM | REDDITB | REDDIT12 |
|---|---|---|---|
| GCNet | $55.9 \pm 4.4$ | $93.2 \pm 1.5$ | $49.1 \pm 1.5$ |
| TeleGCN-$t_s$ | $56.8 \pm 3.6$ | $94.1 \pm 0.9$ | $49.9 \pm 1.4$ |
| TeleGCN-$t_f$ | $56.4 \pm 3.6$ | $93.5 \pm 1.4$ | $49.9 \pm 1.1$ |
| **TeleGCN** | $\mathbf{56.9 \pm 3.2}$ | $\mathbf{94.3 \pm 0.7}$ | $\mathbf{50.1 \pm 1.3}$ |

with teleport functions significantly outperform GCNet by margins of 1.0%, 1.1%, and 1.0% on IMDB-MULTI, REDDIT-BINARY, and REDDIT-MULTI-12K datasets. The results demonstrate the promising contributions of teleport functions.

## 4.4 COMPARISON WITH GEOM-GCN ON NODE CLASSIFICATION TASKS

In previous sections, we evaluate our methods using graph classification tasks under inductive settings. Here, we conduct experiments to evaluate our methods on node classification tasks. We compare our TeleGCN with GCN, GAT, and Geom-GCN on Chameleon and Squirrel datasets (Rozemberczki et al., 2019). To ensure fair comparisons, we use the same experimental setups as Geom-GCN. The statistics of datasets and results are summarized in Table 4. From

Table 4: Comparison results of our TeleGCNs with GCN, GAT, and Geom-GCN on node classification tasks. We report the node classification accuracies on Chameleon and Squirrel datasets.

|  | Chameleon | Squirrel |
|---|---|---|
| GCN | 28.2 | 24.0 |
| GAT | 42.9 | 30.0 |
| Geom-GCN | 60.9 | 38.1 |
| **TeleGCN** | **62.0** | **39.1** |

the results, we can see that our TeleGCN outperforms previous models by margins of 1.1% and 1.0% on Chameleon and Squirrel datasets, respectively. This demonstrates the superior performances of our TeleGCN over previous state-of-the-art models.

## 4.5 PERFORMANCE STUDIES OF THRESHOLDS IN TELEPORT FUNCTIONS

In our proposed TeleGCL, teleport functions employ thresholds to determine if the relevance of nodes is significant. Thus, the threshold is a very important hyper-parameter in our proposed methods. It controls the number of nodes teleported in the feature aggregation process. In this section, we conduct experiments to investigate how different threshold values affect the performance of TeleGCN models. In our TeleGCNs, all teleport functions share the same threshold that is $\alpha/|V|$. This can help to accommodate input graphs with variable sizes. In the experiments of previous sections, we set the hyper-parameter $\alpha$ to 2. Here, we vary the thresholds in TeleGCNs and evaluate the resulting networks on D&D, PROTEINS, and IMDB-MULTI datasets. We report the graph classification accuracies (%) as illustrated in

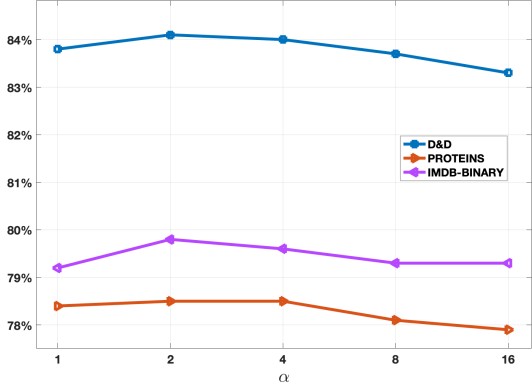

Figure 4: Comparison of TeleGCNs with teleport functions of different thresholds on PTC, PROTEINS, and REDDIT-BINARY datasets.

Figure 4. As demonstrated in the figure, the model achieves the best performance when $\alpha = 2$. When the threshold is small, many nodes are teleported for feature aggregation, thereby leading to an over-smoothing problem. As the threshold increases, fewer nodes are selected and the receptive field is not effectively enlarged.

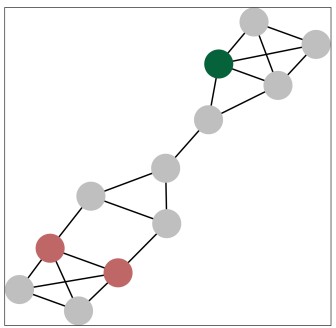 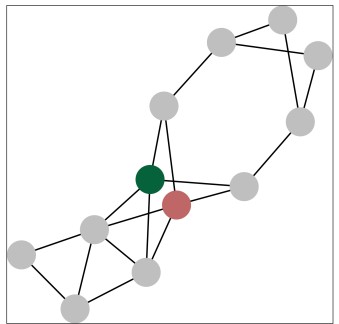 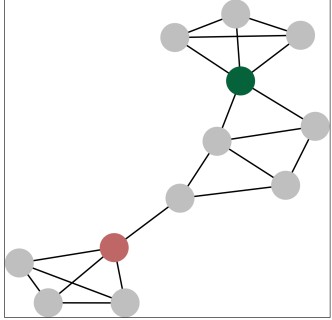

Figure 5: Visualization of teleported nodes. In these graphs, green nodes are center nodes. Yellow nodes are teleported to center nodes by our structure-aware teleport function. From these graphs, we can see that the teleported nodes (yellow) share similar structural patterns as center nodes.

### 4.6 PERFORMANCE STUDIES OF $k$

In our structure-aware teleport function, we propose a novel method to construct structural features for teleport function to compute similarities of nodes from graph structural perspective. Essentially, each node uses the connections with $k$ neighborhoods to build a $k$-dimensional feature vector. From this point, $k$ is an important hyper-parameter especially for our TeleGCL. In this section, we conduct experiments to study the impacts of different $k$ values

Table 5: Comparison of our TeleGCNs using different $k$ values when constructing structural features in structure-aware teleport functions. We report the graph classification accuracies.

| $k$ | D&D | PROTEINS | IMDBB |
|---|---|---|---|
| 2 | 83.8 | 78.6 | 79.2 |
| 4 | 84.0 | 79.0 | 79.5 |
| 8 | 84.1 | 79.0 | 79.8 |
| 16 | 84.1 | 78.8 | 79.8 |

on overall model performances. To this end, we vary the values of $k$ in TeleGCLs and evaluate the resulting models on three datasets; those are D&D, PROTEINS, and IMDB-BINARY. We report graph classification performances on these datasets. The results are summarized in Table 5. We can observe from the results that the networks achieve the best performances when $k = 8$. As the increase of $k$, there is no significant improvement in network performances but the computational cost for computing relevances will increase. Thus, we set $k = 8$ in our experiments as it is the best practice for both efficiency and performance.

### 4.7 VISUALIZATION OF TELEPORTED NODES

In our proposed structure-aware teleport function, the nodes that share the same structural patterns as the center node are teleported. In this part, we provide some visualization analysis of these teleported nodes. Here, we select three graphs from the PROTEINS dataset and visualize them in Figure 5. The green node in each graph is the center nodes and the orange nodes are teleported by the structure-aware teleport function. We can observe from these graphs that the teleported nodes share very similar graph topology information to their corresponding center nodes. The teleported nodes in the first and third graphs are multiple hops away from the center nodes. The teleported nodes enable the center nodes to aggregate information from a larger receptive field. This demonstrates that our proposed structure-aware teleport function can select informative nodes for center nodes beyond the local neighborhood.

## 5 CONCLUSION

In this work, we address two major limitations of graph convolutional networks that are usually based on message-passing operations. To overcome the over-fitting and over-smoothing issues, we propose the teleport graph convolution layer, which utilizes teleport functions to select relevant nodes beyond the original graph structure. In particular, we propose two teleport functions; those are structure-aware and feature-aware teleport functions. These two teleport functions can select relevant nodes from structural and feature perspectives. Based on our TeleGCL, we construct teleport graph convolutional networks on network embedding learning.

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

## A   APPENDIX

In this section, we introduce the experimental setup for our experiments. In this work, we mainly utilize graph classification tasks to demonstrate the effectiveness of our proposed methods. We conduct experiments on ten datasets; those are PROTEINS, COLLAB, D&D, IMDB-BINARY, IMDB-MULTI, MUTAG, PTC, REDDIT-BINARY, REDDIT-MULTI5K, and REDDIT-MULTI12K. In our proposed TeleGCNs, we use a GCN layer as the initial embedding layer. After that, three blocks as described in Section 3.5 are stacked before the final multi-layer perceptron. In each block, we use a TeleGCL and a gPool layer (Gao & Ji, 2019). The output features of the initial GCN and TeleGCLs are reduced to three 1-dimensional feature vectors using global max, global averaging, and global summation operations. These feature vectors are concatenated and fed into a two-layer perceptron. The number of hidden neurons is 512. In each convolutional layer, we apply a dropout (Srivastava et al., 2014) to the feature matrix. In the multi-layer perceptron, we also use dropout on input feature vectors. The hyper-parameter tuning is performed on the D&D dataset with slight changes on other datasets. The details of hyper-parameters are summarized in Table 6.

Table 6: Hyper-parameters for each dataset used in this work.

|          | tel k | tel thred | layer dim | net drop | cls drop | net act   | cls act |
|----------|-------|-----------|-----------|----------|----------|-----------|---------|
| PROTEINS | 8     | 2         | 64        | 0.3      | 0.3      | ELU       | ELU     |
| COLLAB   | 8     | 2         | 64        | 0.3      | 0.2      | ELU       | ReLU    |
| D&D      | 8     | 2         | 128       | 0.3      | 0.3      | ELU       | ELU     |
| IMDBB    | 8     | 2         | 48        | 0.3      | 0.2      | LeakyReLU | ELU     |
| IMDBM    | 8     | 2         | 48        | 0.1      | 0.1      | LeakyReLU | ELU     |
| MUTAG    | 8     | 2         | 64        | 0.0      | 0.1      | ELU       | ELU     |
| PTC      | 8     | 2         | 64        | 0.2      | 0.2      | ELU       | ELU     |
| REDDITB  | 8     | 2         | 64        | 0.1      | 0.01     | ELU       | ELU     |
| REDDIT5  | 8     | 2         | 64        | 0.0      | 0.01     | ELU       | ELU     |
| REDDIT12 | 8     | 2         | 64        | 0.0      | 0.01     | ELU       | ELU     |

On each dataset, we run experiments for 200 epochs by using an Adam optimizer (Kingma & Ba, 2015). The learning rate is 0.001 with a weight decay of 0.0008 to reduce the risk of over-fitting. The batch size is set to 64. We use a NVIDIA GeForce RTX 2080 Ti GPU to train our models.

