# OpenReview forum: "Teleport Graph Convolutional Networks"
_ICLR.cc/2021/Conference — Reject_

### Official Review · AnonReviewer1 · 2020-10-13
**# Official Review of Teleport Graph Convolution Net**

**Rating:** 5
**Confidence:** 4

**Review:**

A new architecture for graph neural networks, which the authors name as Teleport Graph Convolutional Networks (TGL), is proposed in this paper.  Teleport graph convolution layer is proposed to address the limitations in message-passing operations of graph neural networks:  1.  over-smoothing and 2. over-fitting. The architecture enables nodes to aggregate information beyond their local neighborhoods. TGL operates as follows: 1) it first aggregates neighbors as in normal graph neural networks, 2) it selects some nodes that are relevant but not directly connected using 2a) feature similarity and 2b) similarity of structure.

Specifically, TGL builds upon Pei et al., 2020, which attacks the over-smoothing problem by mapping node features to latent spaces and performing aggregation based on latent space similarity. Authors motivate their work by reporting Pei et al., 2020 ignore the graph topology, but TGL can incorporate graph topology with a similar latent space approach. Experiments are conducted on graph and node classification datasets.

Overall, the paper presents some useful attempt to address problems in message passing operations, and some accuracy gains are shown in the graph and node classification datasets. The trick to aggregate based on similarity of features and structure leads to some gains compared to those only based on graph topology. However, there are some concerns lingering around.

I. The presented idea, though boosts accuracy of state-of-the-art graph neural networks such as GIN or Geom-GCN, by 2%, is not quite novel. This paper is perhaps better suited for more practical venues, while ICLR is reserved for the most exciting and eye-opening research in AI.

II. How does teleGCN solve over-smoothing, which is supposed to be the main problem? Not discussed in paper.

III. The motivation for how structure-aware teleport is constructed is unclear yet. Why is yellow node similar to green node in Figure 1? How are their similarity computed in equations, which is missing in paper?

IV. How is the feature-aware (pink node in Figure 1) different from Pei et al., 2020?

V. I recommend authors to include more state-of-the-art baselines for comparison. The following are recommended.
https://arxiv.org/abs/2002.05287
https://arxiv.org/abs/1905.13192
https://arxiv.org/pdf/2007.02133.pdf
https://arxiv.org/abs/2006.07739


Grammar / typos:

Geom-GCN doesn’t consider the -> does not

those are structure-aware teleport function  -> those that are

These similarity scores encode its connectivity -> "its" is unclear

---

### Official Review · AnonReviewer3 · 2020-10-24
**Review - AnonReviewer3**

**Rating:** 5
**Confidence:** 4

**Review:**


**Summary:**

The paper proposes a method to increase the receptive field of GNNs, while avoiding oversmoothing. The idea is to create extra connections by linking distant nodes based on two criteria: node feature similarity and node structure similarity. Pairs of nodes that are more similar than a threshold are connected. For structure-aware linking (teleport) a descriptor of the local structure is constructed for each node, by stacking the similarity scores to the most similar k neighbours. Computing the dot product between descriptors, gives a similarity measure of the local structure of each node. For feature-aware linking the features dot product is directly used as a measure. Experiments show that both types of teleport functions are helpful.


**Strong Points:**

Clearly motivated and simple method. Carefully adding extra connections in the graph is a good idea, and the two proposed ways are sound.

The paper highlights the importance of extra connections based on structure similarity and proposes an interesting mechanism to achieve this.

The visualizations in Figure 5 are useful.

**Weak Points:**

There are some concerns regarding the evaluation of the method.
 a)  The comparing methods from Table 1 have poorer performance than the GCNet baseline used in this work. The baseline GCNet represents a simple GNN formed by stacking GCN and gPool layers and Table 3 shows that it has better results than all the comparing methods from Table 1. Compared to the gap to other methods, the performance improvement brought by the proposed TeleGCL is relatively minor.

 b) The performance for GCNet baseline should be given for every dataset. Since on graph classification the only other results are from the Geom-GCN paper, it would really help to present the results of GCNet baseline on these datasets, with the same splits and settings as in the Geom-GCN paper.

More details about the computational complexity of the model should be given. Due to the computation of the pairwise dot product between all the nodes, the feature-aware teleport function has complexity $O( n \times n \times d )$. Similarly,  the structure-aware function has complexity $O( n \times n \times k  + n \times n \times d)$ (or $|E| \times d$ for the second term, depending on the implementation). For large graphs, the quadratic term in the number of nodes is really problematic. This should be compared to the approach of Geom-GCN.

Some statements regarding Geom-GCN must be clarified: “ Geom-GCN doesn’t consider the
original graph topology information when generating the additional set of nodes for aggregation” or “structural neighborhood in (Pei et al., 2020) is still built on node features without considering the graph topology”. Like the current method, Geom-GCN creates links between nodes that are similar in an embedding space. Their embeddings are given by  Isomap (Tenenbaum et al., 2000), Poincare embedding (Nickel & Kiela, 2017), and struc2vec (Ribeiro et al., 2017) and it seems that they also take into account the structure of the graph. The proposed method given in Eq. 3-5 could be a good alternative to these methods, but more detailed explanations and comparisons should be given.


**Additional Comment:**

Fair evaluation of graph methods on small datasets is challenging. It would have been better to evaluate the method on bigger datasets like OGB [A].


[A] Hu, Weihua, et al. "Open graph benchmark: Datasets for machine learning on graphs." arXiv preprint arXiv:2005.00687 (2020).


**Conclusion**:

This paper presents a good and clear idea to add additional connections in a graph, but it has some issues with the evaluation and some comparisons with prior work. In this form I am inclined towards giving a  *5: marginally below* rating.

---

### Official Review · AnonReviewer4 · 2020-10-28
**The manuscript proposes to use teleport functions to compute node embeddings considering a larger receptive field. Questionable hyper-parameter validation strategy. Missing several references to related works.**

**Rating:** 3
**Confidence:** 3

**Review:**

The manuscript proposes a novel convolutional layer that computes a node embedding by selectively summing the nodes' embeddings neighbors at different hop distances (from 1 to m hops).  The aim of the Teleport graph convolutional layer is to solve the major limitations introduced by using the message-passing paradigm.

In the introduction, the authors highlight the main issues related to message passing paradigms and deep GNN models. The authors state that stacking multiple layers involves massive trainable parameters, which consequently increases the risk of over-fitting.  It is important to note that the issue of having several different GNN stacked layers were already faced in some papers: “The Graph Neural Network Model” by Scarselly et al. (2019), and in “Gated Graph Sequence Neural Networks” by Li et al. (2016). In these papers, the authors use recurrent models that exploit the weights sharing mechanism in order to limit the number of trainable parameters. Moreover, several works that propose an alternative to the message passing paradigm were published in the last few years. For instance, several convolutional operators are designed to consider a larger receptive field. By exploiting power series of the diffusion operator or leveraging on a multi-scale operator:  Diffusion-convolutional neural networks, Atwood et al. (2016), LanczosNet: Multi-scale deep graph convolutional networks, Liao et al. (2019), Break the ceiling: Stronger multi-scale deep graph convolutional networks, Luan et al. (2019), SIGN: Scalable Inception Graph Neural Networks, Rossi et al. (2020). Since the Teleport GCN tries to solve the same issues, all these models should be discussed in the related works section, and also considered in the comparison of the experimental results.
In this regards note that, even if the idea of using the features-aware or the structure-aware teleport function is interesting, the teleport convolutional layer defined in section 3.2 is not very novel in my opinion, since it seems very close to a multi-scale approach where in the same layer several exponentiations of linear diffusion operator are considered.

The main problem of this work is the empirical evaluation of the proposed method. The authors validate the models using an unfair method. Indeed the authors state “ The hyper-parameters in TeleGCNs are slightly turned on D&D dataset and are migrated to other datasets with slightly different selections' '. Using similar hyper-parameters tuned in D&D for the other datasets in my opinion is not correct. Note that the other datasets differ significantly from D&D (e.g. PROTEINS has 39.1 nodes vs 284.3; COLLAB has a significantly different number of classes and 5 times the number of graphs; etc.). It is also important to notice that all these datasets require a different type of classification task.
Moreover, the method used by the authors differs from the “common practice”  used in (Xu et al., 2018; Ying et al., 2018; Gao & Ji, 2019; Lee et al., 2019) since in Xu et al., 2018 the authors state: “The hyper-parameters we tune for each dataset are: (1) the number of hidden units ∈ {16, 32} for bioinformatics graphs and 64 for social graphs; (2) the batch size ∈ {32, 128}; (3) the dropout ratio ∈ {0, 0.5} after the dense layer (Srivastava et al., 2014); (4) the number of epochs, i.e., a single epoch with the best cross-validation accuracy averaged over the 10 folds was selected”. The validation phase has to be executed independently for each dataset, otherwise, the obtained result will be clearly biased. Moreover, note that the impact of the validation phase in evaluating the performance of a model is discussed in “A Fair Comparison of Graph Neural Networks for Graph Classification” by Errica et al. (ICLR 2019). The results reported in this paper show that performing a fair validation procedure is crucial to evaluate the model's performance.

Minor comments:
From section 4.2 to section 4.7  it is almost impossible to distinguish between tables' captions and sections text.

---

### Official Review · AnonReviewer2 · 2020-10-29
**This paper provided two novel teleport functions for graph convolutional networks, which enables each node to aggregate information from a larger neighborhood. Specifically, it proposed to choose the neighbors based on structure-aware and feature-aware relatedness between one center nodes and all the potential neighbors from the graph. The experiments demonstrated its superior performance over existing graph convolutional networks.**

**Rating:** 5
**Confidence:** 5

**Review:**

This paper analyzed the key issues of the existing message-passing graph convolutional networks. That is, the multiply stacked layer might be over-fitting and over-smoothing. Thus it proposed to choose the neighbors from the entire graph based on the structure-aware and feature-aware relatedness rather than simply choosing the local neighborhood. However, the motivations of the proposed structure-aware and feature-aware teleport functions are not very convincing, and Table 3 shows that the performance improvement of TeleGCN might largely be induced by model architecture rather than the proposed TeleGCL.

Pros:
[1] It analyzed the potential issues of message-passing operations in conventional graph convolutional networks.
[2] It proposed structure-aware and feature-aware teleport functions to select the neighbors from the entire graph.
[3] The experiments show that the proposed TeleGCN outperforms the existing graph convolutional networks.
Cons:
[1] It is confusing why the proposed TeleGCN model could avoid the risk of over-fitting. From Figure 3, it also stacks multiple teleport graph convolution layers.
[2] Another concern is the scalability of the proposed graph convolutional networks. It might not be efficient to be applied to large-scale networks by selecting the neighbors from the entire graph for every TeleGCL. Besides, it might be more convincing to empirically compare the running time of the proposed method to the baselines.
[3] For structure-aware teleport function, it involved the top-k similar connected neighbors as the structural feature vector. Then why not consider the high-order structural similarity or specific graph motif similarity?
[4] From Table 3, it shows that the performance improvement of TeleGCL is very limited and the GCNet model can obtain superior performance. That might indicate that the performance improvement of TeleGCN might largely be induced by model architecture rather than the proposed TeleGCL.
[5] It is not clear which papers the baselines, e.g., WL, PSCN, DIFFPOOL, etc., are from.

---

### Decision · Program_Chairs · 2021-01-07
**Final Decision**

**Decision:**

Reject

**Comment:**

The paper seeks to increase receptive fields of GNNs by aggregating information beyond local neighborhoods with the idea of addressing oversmoothing and/or overfitting issues with message passing algorithms. The proposed method is simple and primarily makes use of node features and local structure similarities. In this sense the approach is related to Pei et al. Several concerns remained as articulated in the reviews, including: oversmoothing is not discussed/analyzed, performance gains are small, more extensive comparisons are needed.